# Scoping Review of Climate Change and Health Research in the Philippines: A Complementary Tool in Research Agenda-Setting

**DOI:** 10.3390/ijerph16142624

**Published:** 2019-07-23

**Authors:** Paul Lester Chua, Miguel Manuel Dorotan, Jemar Anne Sigua, Rafael Deo Estanislao, Masahiro Hashizume, Miguel Antonio Salazar

**Affiliations:** 1Alliance for Improving Health Outcomes, Inc., Rm. 406, Veria I Bldg., 62 West Avenue, Barangay West Triangle, Quezon City 1104, Philippines; 2Department of Global Health, School of Tropical Medicine and Global Health, Nagasaki University, 1-12-4 Sakamoto, Nagasaki 852-8102, Japan; 3Department of Pediatric Infectious Diseases, Institute of Tropical Medicine, Nagasaki University, 1-12-4 Sakamoto, Nagasaki 852-8523, Japan; 4Institute of Global Health, University of Heidelberg, Im Neuenheimer Feld 324, 69120 Heidelberg, Germany

**Keywords:** climate change, health, scoping review, agenda-setting

## Abstract

The impacts of climate change on human health have been observed and projected in the Philippines as vector-borne and heat-related diseases have and continue to increase. As a response, the Philippine government has given priority to climate change and health as one of the main research funding topics. To guide in identifying more specific research topics, a scoping review was done to complement the agenda-setting process by mapping out the extent of climate change and health research done in the country. Research articles and grey literature published from 1980 to 2017 were searched from online databases and search engines, and a total of 34 quantitative studies were selected. Fifty-three percent of the health topics studied were about mosquito-borne diseases, particularly dengue fever. Seventy-nine percent of the studies reported evidence of positive associations between climate factors and health outcomes. Recommended broad research themes for funding were health vulnerability, health adaptation, and co-benefits. Other notable recommendations were the development of open data and reproducible modeling schemes. In conclusion, the scoping review was useful in providing a background for research agenda-setting; however, additional analyses or consultations should be complementary for added depth.

## 1. Introduction

The Philippines is one of the most vulnerable nations where one can observe and project the impacts of climate change [1]. Climate change-induced temperature increases and rainfall variability are considered most likely to have the greatest impacts on the country [1]. The frequency and intensity of tropical cyclones originating in the Pacific are also increasing [2], albeit not definitive [3], and three of the highest recorded maximum gustiness occurred in this country in the last two decades, including Typhoon Haiyan. Studies found that climate change will continue to expose the vulnerabilities of the ecosystems, freshwater resources, coastal systems, agriculture, and fisheries in the Philippines, as well as human health. It was observed and projected that climate change affected, and will continue to affect increases in diseases—particularly vector- and waterborne diseases—as well as heat-related illnesses [1].

In 2015, the Philippine Government spent a total of 2.83 million USD for climate change adaptation (98%, 2.73 million USD) and mitigation (2%, 0.063 million USD) [4]. These were allocated to food security (3%), water sufficiency (54%), the ecology and environment (8%), human security (5%), climate-smart industries (0.4%), sustainable energy (26%), knowledge and capacity development (1.8%), and finance (0.8%) [4]. None had been allocated for health or spent under the health sector.

The lack of attention to the impacts of climate change on health was picked up by the Philippine National Health Research System (PNHRS), who included it in the National Unified Health Research Agenda 2017–2022 as one of the core topics under the Health Resiliency section [5]. To help map out and elaborate the current extent of this research field, a research agenda setting was conducted in 2018. As part of the agenda-setting process, a scoping review was done to help elaborate on and generate possible research themes and priorities for funding, and this paper presents the results of the review.

Since scoping reviews can be an arduous, yet transparent method for mapping research areas, some recently used it as either a stand-alone or supplementary tool in agenda-setting in various fields in health research [6,7,8,9]. Montesanti et al. used it as a sole tool in generating key research themes in primary health care in Canada to contribute to the efficient and equitable use of limited funding, and possibly reduced duplication [6]. Alternatively, Ajumobi et al. used a scoping review in developing introductory materials to guide the consultative process in setting Nigeria’s national malaria operational research agenda [7].

In the field of climate change and health, Hosking and Campbell-Lendrum used a scoping review to generate an overview of the entire field of climate change and health guided by the World Health Assembly priorities as a framework, including: (1) assessing risks, (2) identifying effective and cost-effective interventions, (3) measuring the co-benefits and co-harms of adaptation and mitigation, (4) improving decision support, and (5) estimating costs [10]. Recently, a more rigorous scoping review methodological framework for climate change and health was developed by Herlihy et al. to examine historical trends and provide a more extensive and inclusive overview of existing scientific literature on climate change and health based on the Intergovernmental Panel on Climate Change (IPCC) framework [11].

Similarly, the purpose of this paper is to present an overview of the existing literature on climate change and health research in the Philippines. The specific objectives of this review are to: (1) map out the health topics associated with climate change in the Philippines, (2) describe their methodologies, findings, and reported challenges, and (3) identify possible research priorities.

## 2. Materials and Methods

The methodology used was based on the approach outlined by Arksey and O’Malley, and search strategies by Herlihy et al. [11,12]. The review comprised: (1) identifying a broad research question, (2) identifying relevant studies, (3) selecting studies, (4) charting the data, (5) collating, summarizing, and reporting the results, and (6) consulting stakeholders. The research team came up with the broad question, “What are the characteristics, methodologies, and findings of the studies on climate/weather exposure and human health outcomes done in the Philippines?”

### 2.1. Search Strategy

The literature search was limited to online-based databases or search engines. Initially, a literature search was done using PubMed/MEDLINE, Embase, and the Web of Science on March 7, 2018. For grey literature, additional online databases and search engines were used, including the Health Research and Development Information Network (HERDIN), which is a research database of the Philippine Council for Health Research and Development, as well as OpenGrey, ProQuest, and Google on April 5–6, 2018. Grey literature were searched to expand the possibly small number of published articles in scientific journals. The search terms comprised broad terms on climate factors and health outcomes (Table 1) [10,11].

The literature search was limited from 1980 to 2017 (37 years) to possibly consider studies in the past. References of screened full-text articles and grey literature were manually searched for additional literature.

### 2.2. Eligibility Criteria

For published articles, only full-text original or research articles were selected, while grey literature were limited to full-text theses, dissertations, technical reports, and discussion papers. Clear statements of associations between climate factors and health outcomes were required. Climate factors were meteorological parameters, extreme weather events, air pollution, and atmosphere-ocean interaction phenomena. The Philippines had to be explicitly mentioned as a study site. In the case of multi-site studies, findings specifically for the Philippines should be present in the results or discussion section. Initially, both quantitative and qualitative studies were considered to further widen the selection pool of articles and papers; however, quantitative studies were only presented here to clearly articulate the results. Mixing the results and discussion with qualitative and/or mixed methods studies may limit the depth of the discussion. Only articles and grey literature written in English were included, because it is the language used in academia or business in the Philippines.

### 2.3. Screening

The abstracts and executive summaries were reviewed by two individuals looking for the keywords related to climate factors, human health outcomes, associations between exposures and outcomes, and the Philippines as the study site or one of the study sites. Studies with Filipino participants done outside the country were excluded (e.g., studies with overseas Filipino workers as participants). After the initial screening, each reviewer separately reviewed the full texts of the articles and papers. An eligibility criteria review was done to examine their content in terms of associations made between health outcomes and climate factors in the Philippines. For the purposes of this study, an association was loosely defined as any explicit textual statement on the attempt to relate health outcome(s) with climate factor(s). The results of the screening were discussed by two reviewers to reach a consensus on which should be included in the final selection list.

### 2.4. Characterization

The selected articles and papers were categorized based on the human health outcomes, climate factors, research design, methodologies, and country affiliations of the first authors which were researched. Further data on the study results were extracted to identify the specific details on the type of research and depth of the findings. To identify research themes, five climate and health themes by Smith et al. in one of the chapters of the Intergovernmental Panel on Climate Change Fifth Assessment Report were adopted [13]. Apart from the purposes of brevity, this was selected because the eligibility criteria followed Smith et al.’s concept of linking primary exposure pathways by which climate change affects health outcomes. To guide the categorization of the articles, excerpts from their study objectives and results were used as a basis from which to identify which theme they fell under (Table 2).

### 2.5. Consultation

Initial results of the review were presented to 20 stakeholders from various offices in the Philippines with initiatives or interests in climate change and health, including government agencies, academia, and non-profit institutions, on May 16, 2018. Recommendations and feedback were sought. Some government institutions shared their studies for possible inclusion in the analysis.

### 2.6. Data Analysis

Data were encoded in a single Microsoft Excel 2013 spreadsheet (Microsoft Corporation, Redmond, WA, USA). Descriptive statistics were calculated for summary using MS Excel 2013. Excerpts from selected studies were coded by one author and validated by another.

### 2.7. Ethical Considerations

Since the lead institution (Alliance for Improving Health Outcomes, Inc) did not have an ethical review committee, the research protocol received ethical approval from the St. Cabrini Medical Center, Asian Eye Institute Ethics Review Committee (Makati City, Philippines), Protocol No. 2018-006.

## 3. Results

After removing duplicates, a total of 757 studies (726 articles and 31 grey literature) were retrieved and initially reviewed for their abstracts. A majority (678 articles) were excluded since they were non-human health-related. Four grey literature found from HERDIN did not have executive summaries, and were not retrievable upon request from the authors. A total of 75 passed the abstract/executive summary review (Figure 1). Upon full-text review, 16 studies were found to have no mention or analysis about climate factors and findings specifically for the Philippines. Fourteen had no retrievable full-text copies despite requesting from authors. Five were non-research studies. One article was retrieved from references of a published article. Additional three grey literature were shared by consultation participants, and only one passed the full-text review because the other two were about disaster risk reduction. A total of 40 articles and grey literature were found eligible for this review. Only 34 quantitative studies were presented here [14,15,16,17,18,19,20,21,22,23,24,25,26,27,28,29,30,31,32,33,34,35,36,37,38,39,40,41,42,43,44,45,46,47].

### 3.1. General Characteristics

Seventy-nine percent (79%, 27/34) of the selected quantitative studies were articles published in scientific journals (Table 3). Others were technical reports (9%, 3/34), theses/dissertations (6%, 2/34), a discussion article, and a conference paper. Sixty-eight percent (68%, 23/34) of the selected articles and papers were published/dated from 2011 to 2017, while only 11 were published in the previous decades. Health topics tackled in these studies were varied, but about half were on mosquito-borne diseases (53%, 18/34), mostly on dengue (50%, 17/34), and only two on malaria. Some were water-related (26%, 9/34), such as diarrheal diseases, helminth infections, and leptospirosis. Other communicable diseases that were studied were meningitis (6%, 2/34) and measles (3%, 1/34). Four (12%) were related to mortality due to non-communicable diseases, such as cardiovascular diseases and diabetes. One (3%) studied malnutrition in children. Spatial resolutions of the selected studies were mostly city-level (47%, 16/34), and there were least on a region-level (15%, 5/34). Fifty-nine percent (59%, 20/34) of the institutional affiliations of the first authors were from Philippine institutions, while the remaining were from other countries, such as Japan and the US. Eighteen of the studies (53%) were funded by foreign institutions, while only two (6%) received funding from Philippine agencies.

### 3.2. Methodologies

Most of the studies (79%, 27/34) used time-series analysis as the research design in associating exposure variables and health outcomes, while the remaining used cross-sectional (9%, 3/34), process-based modeling (6%, 2/34), Bayesian modeling (3%, 1/34), and fuzzy association rule mining (FARM) (3%, 1/34) (Table 4). A time-series study also applied a case-crossover design, which used case days (two control days per case day).

Among the time-series studies, six (22%) simply visualized patterns from time-series plots [28,31,39,40,43,47], while the rest used a variety of statistical models. Eight (30%) used general linear models (i.e., simple and multiple linear regressions) [27,32,34,35,38,41,44,45], eight (30%) used generalized linear models (e.g., quasi-poisson/poisson models and distributed lag nonlinear models) [15,19,22,25,33,36,42], two (7%) used autoregressive models (i.e., autoregressive integrated moving average models and seasonal autoregressive integrated moving average models) [37,44], two (7%) used wavelet analysis [16,20], and four (15%) used other kinds of models (i.e., general additive model [14,15], spectral analysis [18], dynamic linear model [16], and transfer entropy [46]). The temporal resolutions used in the time-series studies were daily (19%, 5/27), weekly (15%, 4/27), monthly (63%, 17/27), and annual (7%, 2/27). Only 33% of these studies (9/27) considered temporal lags of exposure to health outcomes [14,15,18,22,25,36,37,42,44]. All analyzed short-term associations of disease patterns were on preceding and concurrent exposure, but only seven studies controlled temporal trends and seasonal patterns [14,15,22,25,36,37,42]. Two studies explored longer-term associations with annual data and atmosphere-ocean interaction phenomena. One study using SARIMA forecasted monthly dengue incidence from 2011 to 2014 [37], and a study projected dengue, malaria, and cholera cases to 2020 and 2050 using general linear models [44].

The two process-based modeling studies (i.e., CLIMEX and the susceptible-infected-recovered model) were hybrid models incorporating time-series [17,24]. The three cross-sectional studies used descriptive statistics to present associations between dry/wet seasons and health outcomes. One study applied the Bayesian logistic geostatistical model for schistosomiasis [21]. Only two models forecasted health outcomes (i.e., the CLIMEX model for malaria prevalence and FARM predictor model for dengue incidence). The CLIMEX model projected malaria prevalence up to year 2100 based on eco-physiological suitability for mosquitoes. Similarly, the predictor model purposively predicted dengue outbreaks four weeks in advance based on environmental predictors related to suitability for mosquitoes.

Based on the exposure variables, 27 studies (79%) used various meteorological parameters, five (15%) used extreme weather events, three (9%) used atmosphere-ocean interaction phenomena, and three (9%) used wet and dry seasons. The meteorological parameters used were rainfall (100%, 27/27), air temperature (89%, 24/27), humidity (67%, 18/27), sunshine hours (4%, 1/27), sea-level pressure (4%, 1/27), wind speed (4%, 1/27), and dew-point temperature (4%, 1/27). Most of the studies’ sources (56%, 19/34) of these parameters was the Philippine Atmospheric, Geophysical, and Astronomical Services Administration (PAGASA), which routinely observes these parameters via synoptic weather stations. Other sources were the US National Oceanic and Atmospheric Administration (18%, 6/34), WorldClim (6%, 2/34), CliMond (3%, 1/34), and National Aeronautics and Space Administration-Prediction of Worldwide Energy Resources (3%, 1/34). For rainfall, three studies used counts of rainy days rather than using the total/mean rainfall in millimeters. The extreme weather events used were high temperature days (categorized by high temperature percentiles), heatwaves (consecutive days with high temperature percentiles), and typhoons. For the atmosphere-ocean interaction phenomena, El Niño Southern Oscillation (ENSO) indices (i.e., Southern Oscillation Index, Oceanic Niño Index, and ENSO years) and the sea-surface temperature were used.

In terms of outcome variables, the most common (44%, 15/34) was surveillance data from the Philippine Department of Health and a non-government active surveillance. These were secondary data from the existing surveillance system of selected public hospitals and clinics. Nine studies (26%) used hospital admissions data, in which five were primary and four were secondary medical records. Four studies (12%) using mortality data were from vital statistical records of the Philippine Statistics Authority. Three (8%) used data from government- and foreign-funded national surveys for schistosomiasis, enteric protozoans, and poverty, while others were hospital outpatient-, community-, and school-based human and non-human samples that were collected.

### 3.3. Findings

Eighty-two percent (28/34) of the studies reported associations between exposures and health outcomes, with 27 studies (79%) reporting positive associations [14,15,16,18,20,21,22,24,25,26,27,29,30,31,32,33,34,36,37,38,39,41,42,44,45,47], and six studies (18%) reporting negative associations [16,19,34,36,46,47]. Fifteen studies (44%) reported no or insignificant associations [19,24,25,27,28,33,34,35,37,38,40,41,44,46,47].

For the 15 time-series studies of dengue incidence, 47% (7/15) reported significant positive associations with air temperature (i.e., mean, minimum, and/or maximum) [18,33,34,38,41,42,47], while only two reported a negative association with maximum temperature [34,44]. By selecting only El Niño years in 1997–1999, Naragdao (2001) found that the mean air temperature associations had higher effect to dengue incidence in the Iloilo province (from a coefficient of 0.0118 in pre-El Niño years of 1990–1996, to a coefficient 2.0174 in El Niño years of 1997–1999) [42]. For rainfall, 47% (7/15) of the dengue studies reported positive associations [18,27,33,37,42,43,47], while three reported negative associations [19,44,46]. Agustin found that years with La Niña (phenomena with abnormally high rainfall) were positively associated with dengue incidence [41]. Other positive associations with dengue incidence were relative humidity (33%, 5/15) [18,42,47] and maximum sea-level pressure (7%, 1/15) [33]. For a study on dengue virus samples, its minimum infection rate was found to be positively associated with relative humidity [19,37,38,44,46]. On the other hand, studies reported insignificant associations (*p* > 0.05) between dengue incidence and relative humidity (40%, 6/15) [19,33,35,37,38,41], rainfall (27%, 4/15) [34,35,38,41], air temperature (20%, 3/15) [19,27,35], and minimum sea-level pressure (7%, 1/15) [33].

Two studies on malaria incidence revealed associations with air temperature and rainfall. For air temperature, positive associations were found with maximum and mean temperatures [44,47]. However, Lorenzo et al. found a negative association with total monthly rainfall [44], while PAGASA found a positive association with total monthly rainfall. Both found insignificant associations between malaria incidence and relative humidity [47].

For communicable respiratory diseases, interesting associations with exposures were found. Acute respiratory infection prevalence was positively associated with wind speed, but insignificantly associated with relative humidity, mean temperature, and accumulated rainfall [45]. Pneumonia incidence in children less than three years old was negatively associated with sunshine hours with a cumulative 60 day lag (relative risk (RR) of 0.67 (95% confidence intervals (CI): 0.52–0.87)), but not associated with rainy days, relative humidity, and mean temperature with a cumulative 60 day lag [25]. For influenza and the respiratory syncytial virus (RSV), positive associations were found with mean temperature and specific humidity, but negative associations were found with relative humidity and precipitation [24]. Both influenza and RSV had no association with the number of rainy days per week [24].

In water-related diseases, three studies observed that diarrhea incidence would peak during the rainy season, particularly the months with highest total rainfall [26,30,39]. Additionally, three studies on cholera incidence revealed positive associations with monthly rainfall and mean monthly relative humidity [39,44,47]. In addition, a study found a negative association with maximum temperature [44]. One study on leptospirosis using spectral analysis found a positive correlation between least-square-fitting curves of leptospirosis with rainfall, but an insignificant association with mean temperature [18]. On the other hand, acute bloody diarrhea and typhoid fever incidence was found to be unassociated with mean temperature, rainfall, and relative humidity [46]. For soil-transmitted helminthiases, the land surface temperature was positively associated with prevalence odds of *Ascaris lumbricoides* and *Trichuris trichiura*, and negatively associated with the prevalence of hookworm infections [21]. Additionally, *T. trichiura* prevalence odds was found to be positively associated with rainfall [21].

For other communicable diseases, one study observed that meningitis occurred more during the dry-cool season [29]. Other studies on measles and meningitis found insignificant associations with mean temperature and rainfall [47]. For malnutrition, Salvacion observed high and low total rainfall months to be associated with rates of malnourished children under five years old [32]. In the summer months, the high mean temperature months were found to be negatively associated with the rate of malnourished children [32].

Four studies by Seposo et al. (2015, 2016, 2017, and 2017) explored the effects of extreme heat and heatwaves to mortality (i.e., due to diabetes and cardiovascular, respiratory, and multiple causes) in selected cities [14,15,22,36]. They found that extreme temperatures (i.e., 95th and 99th percentiles of mean temperature) with a cumulative 15 day lag led to high risks in mortality [pooled RR of 2.48 (95% CI: 1.55–3.98)] [15]. The minimum mortality temperature was observed at the 75th percentile of mean temperature (i.e., ~28 °C) [15]. However, the temperature-mortality relationships showed a U-shaped pattern with elevated risks at the extremes of the temperature range [15]. For example, the first temperature percentile RR was 1.23 (95% CI: 0.88–1.72) [15]. They also found that higher risks were particularly observed in respiratory mortality, women, and people aged >64 years [15]. In the city of Manila (capital of the Philippines), the effect modification of age in the 99th temperature percentile-mortality relationship were increasing with RRs of 1.23 (95% CI: 1.07–1.41) for ages of 0–14 years of age, 1.31 (95% CI: 1.18–1.46) for 15–64 years of age, and 1.53 (95% CI: 1.31–1.80) for >64 years of age [22]. For diabetes mortality, the highest RR with extreme temperature was observed with a shorter cumulative lag of 7 days [RR: 1.61 (95% CI: 1.21–2.15)] compared to 21 days [RR: 1.55 (95% CI: 1.06–2.29)] [14]. On the other hand, the heatwave effect to all-cause mortality was found to be insignificant and mostly negative [36].

For the studies using process-based models, findings were presented in a different manner. Khormi and Kumar observed that the Philippines had suitable to highly suitable conditions (i.e., ecoclimate conditions comprising mosquito growth and stress indices) for the mosquitoes to survive and for malaria transmission in the reference years of 1950–2000 [17]. Based on ecoclimate conditions in 2100, projections show an overall reduction in the climate suitability for *Anopheles* in the Philippines because of changes in heat stress, causing large areas to have heat stress beyond the maximum survival values for the malaria vector (>40°C) [17]. Using the susceptible-infected-recovered (compartmental) model, Paynter et al. modeled the seasonal peak and troughs of respiratory syncytial virus transmission [24]. They found that rainfall seasonal troughs occurred consistently post 17–18 weeks after the centre of yearly RSV epidemics [24]. However, they did not see such a lag pattern when visually compared to relative humidity, dew point, and mean temperature [24]. For the Buczak et al.’s prediction model using FARM, findings reported were about the model’s considerably high accuracy in predicting weekly dengue incidence in Philippines provinces four weeks in advance [23]. They explained that creating rules/fuzzy sets of meteorological and climate parameters based on existing literature was enough to predict dengue outbreaks [23].

### 3.4. Challenges and Recommendations

Majority of the studies (62%, 21/34) did not report limitations related to climate change and health [18,19,23,24,25,26,27,28,29,30,31,32,33,35,36,37,38,39,40,41,43,46]. With the ones reporting limitations, recurring statements of limitation were related to poor data quality (35%, 12/34) [14,15,16,17,20,21,22,34,42,44,45,47], as studies reported a lack of long daily time-series data [16,34], significant missing values [22,42,44,45], and unavailable geographical or spatial data [15,16]. Apart from these limitations, other vital environmental parameters like air pollution [14,15,22] and migration [20] were not available to further provide depth in the data analysis. To reduce such limitations, Buczak et al. recommended investing in data quality and monitoring improvements so that disease incidence surveillance and meteorological parameters can be accurate, reliable, and accessible [29].

Further analyses were also recommended: (1) in associating climate/weather variables and health outcomes by using advanced modelling techniques [18,24,27,45,46]; (2) in verifying harvesting, temporary increase in deaths, and effect modification by socioeconomic factors [14,15]; (3) by expanding to other study sites [21,33,44,45]; and (4) in exploring evidence of causality between exposure and health outcomes [25,45].

Frequently recommended use of the study findings was to serve as a basis in improving policy and programmatic implementation (56%, 19/34) [14,15,17,18,19,20,21,22,23,27,31,32,33,35,36,38,41,42,44]. An example was the suggestion of Cabrera (1985) in delivering deworming tablets twice a year because of the seasonality of reinfection of helminths in children, which was determined during low rainfall months [31]. Another example was related to intensified vector control for dengue (21%, 7/34), especially during the rainy season or months with high rainfall [19,20,27,33,35,38,41]. Predictive models were also deemed useful as part of an early warning system for disease prevention and control programmes [14,15,19,22,23] or for extreme weather events like heat waves [36]. Furthermore, Naragdao suggested the need for multidisciplinary collaboration across sectors due to the complexity of climate change impacts [42]. Seposo et al. found that the regulation of room temperature for individuals with non-communicable diseases like cardiovascular diseases and diabetes should be explored to reduce risk during summer days with extreme temperature [36].

### 3.5. Research Themes

Sixty-eight percent (23/34) of the studies were under the theme of “Ecosystem-Mediated Impacts of Climate Change on Health Outcomes” (Table 5), with topics related to vector-borne and water-related diseases. The next theme was “Vulnerability to Disease and Injury Due to Climate Variability and Climate Change” (12%, 4/34) with studies about the vulnerability of age groups and gender. The rest of the studies were categorized under “Direct Impacts of Climate and Weather on Health” (32%, 11/34), “Adaptation to Protect Health” (12%, 4/34), and “Health Impacts Heavily Mediated through Human Institutions” (3%, 1/34). None were categorized under “Co-Benefits”.

## 4. Discussion

This study presented the overview of climate change and health studies done in the Philippines. Selected research articles and grey literature were descriptively analyzed to summarize their general characteristics, methods, findings, limitations, and recommendations, as well as to categorize their research themes. Based on the results, recommended research priorities for funding, limitations of the review, and pointers about research agenda setting are discussed as follows.

### 4.1. Research Gaps and Priorities

There were actually not enough studies retrieved, thus making the identification of research gaps not as straightforward as it could be. When looking at the research themes with the least number of studies, there were apparent gaps in research topics on the “Vulnerability to Disease and Injury Due to Climate Variability and Climate Change”, “Adaptation to Protect Health”, “Health Impacts Heavily Mediated through Human Institutions”, and “Co-Benefits”. These research topics remain broad but can potentially produce valuable findings and offer key solutions to several existing problems. For a country with considerable vulnerability to extreme weather events and climate change, prioritizing health vulnerabilities and health adaptation research is an easy recommendation because it potentially addresses present and near-future issues [48]. These can also generate tangible evidence and technology that can be relatively appreciated by policy-makers and the general public.

Another topic that can potentially be appreciated by the general public is co-benefits research. It is a good priority topic for funding, as such studies can quantify co-health benefits in reducing greenhouse gas emissions from major sources like transportation, food and agriculture, energy generation, and industries [49]. This research also touched on air pollution research, such as measurements of the impacts of climate-altering pollutants on health, which were not retrieved from the literature search. These can pave the way for creating and supporting mitigation policies and legislations across non-health sectors [49]. The lack of co-benefit studies retrieved may be explained by the complex interdisciplinary approach required from sectors who do not usually collaborate with one another. Thus, apart from providing research funding, workshops and events for establishing climate change research collaborations may be considered to be organized in the future to initiate an exchange of ideas.

A recurring study limitation was data unavailability and poor quality. Local sources of secondary data on health outcomes and meteorological parameters were reported to have limitations on their quality, accuracy, and reliability. These sources are also not openly available, and have limitations due to the current data privacy policies. A possible reason behind this is that these data sources are not made to cater for modeling studies. Since many time-series analysis studies rely on such data, better data monitoring and storage should be included as a priority project alongside such research projects. Exploring the possibility of developing open and anonymized health outcomes and climate change data from existing local databases can unlock a revolution of modeling studies.

Apart from developing better databases, a possible research topic that can be prioritized is the development of standardized and reproducible modeling scheme(s) so that studies can be compared and combined. This considers the limitations in the geographical setting and political setup of the Philippines. Beyond modeling or ecological studies, quantifying causality through (nested) cohort studies, although complex and expensive, can be considered to be developed.

### 4.2. Limitations

A number of limitations can be noted from this scoping review. The studies presented only included quantitative studies, although there were six qualitative and mixed-method studies retrieved. These kinds of studies, although small in number, can provide different insights, particularly in regard to the perceptions, habits, and adaptation practices of people. These are data that cannot be clearly captured using quantitative methods.

Also, the selection of quantitative studies excluded disaster risk reduction (DRR) studies. For example, flooding-related studies were not selected because they were mainly related with DRR. In reality, DRR and climate change studies have commonalities and share risk pathways to health outcomes [50]. Exclusion of studies with no measurements of health outcomes may have also averted the inclusion of relevant studies related with climate change.

Indeed, IPCC research themes provided a simple guide in identifying potential research priorities for funding. However, it fails to provide specific health topics to focus on. From the retrieved studies, dengue was the most researched, but this does not necessarily mean that dengue has been studied well without analyzing the quality of the research and evidence found. The descriptive nature of the scoping review limits such analyses. A possibility method of narrowing down the amount of health topics to focus on is by considering the top causes of mortality and morbidity, or what are the current political priorities. Topics that can have implications to climate change can be the selected as the priority topics. In this way, selection of topics to be funded by the government remains valuable and of interest to the government.

The literature search was not exhaustive enough because it was limited to what was available online. Although there were some literature retrieved from the consultation, this undermines other studies from academic institutions and other unrepresented agencies working on climate change research.

### 4.3. Research Agenda-Setting

The full process conducted in the research agenda setting for climate change and health comprised: (1) scoping review, (2) stakeholder analysis, and (3) consultation meetings. The sole purpose of the scoping review in the agenda-setting was to provide the needed background for the stakeholders in guiding their decision-making in selecting the final research priority topics for funding and generating the roadmap for climate change and health research. Although scoping reviews can be a standalone tool in setting research agenda, this was not done because of the possible disconnection with the local stakeholders’ interests, knowledge, and skills. The power in selecting the final research agenda was still with stakeholders and driven by their own individual or group interests. However, overall, the findings from the scoping review allow the stakeholders to see how their interests fit in the current evidence and gaps, thus discouraging any possible redundancies. Furthermore, any disconnections between the stakeholders and scoping review findings can be beneficial for the funding agencies in possibly considering support of the development of human resources for climate change and health through scholarships and training.

## 5. Conclusions

The use of scoping reviews supports the process of research agenda-setting. This particular study provided an overview of the current status of climate and health research in the Philippines, which allowed for identification of certain gaps and possible research topic priorities, such as health vulnerability, health adaptation, and co-benefits. However, the broad results should require additional analysis or a consultation workshop to conclusively select more specific health research topics for funding.

## Figures and Tables

**Figure 1 ijerph-16-02624-f001:**
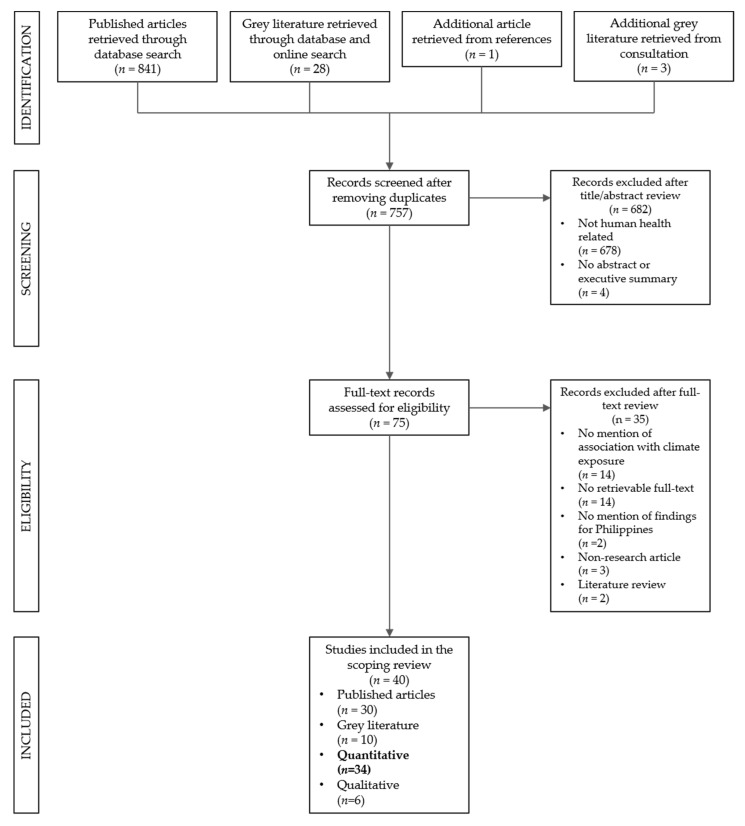
Preferred Reporting Items for Systematic Reviews and Meta-Analyses (PRISMA) flowchart of selection process.

**Table 1 ijerph-16-02624-t001:** Search keywords.

Category	Keywords
Climate factors and health	(“climate” OR “climate change” OR “extreme weather” OR “global warming” OR “climate variability” OR “greenhouse gas” OR “rising temperature” OR “sea-level rising” OR “CO2”) AND (“health” OR “disease” OR “epidemiology” OR “air pollution” OR “water” OR “food” OR “nutrition” OR “vector”)
Location/study site	“Philippines”
Timeframe	“1980” to “2017”

**Table 2 ijerph-16-02624-t002:** Criteria in classifying research themes (adopted from Smith et al. [13]).

Theme	Criteria
Vulnerability to Disease and Injury Due to Climate Variability and Climate Change	Studies that had findings on health vulnerabilities, such as socioeconomic status, age, and gender in relation to the effect of climate change.
Direct Impacts of Climate and Weather on Health	Studies with the aim of and results on associating the direct effects of climate change with health outcomes like mortality and diseases unmediated by the ecosystem.
Ecosystem-Mediated Impacts of Climate Change on Health Outcomes	Studies with the aim of and results on associating climate change with vector-borne, food- and water-borne, as well as air quality-related health outcomes.
Health Impacts Heavily Mediated through Human Institutions	Studies with the aim of and results on associating climate change with nutrition, occupational health, mental health, and violence and conflict.
Adaptation to Protect Health	Studies with the aim of and results on improving basic public health and health care services, as well as the formulation of adaptation policies and measures (including early warning systems) based on associated health impacts of climate change.
Co-benefits	Studies with the aim of and results related to mitigation measures and their benefits to health outcomes.

**Table 3 ijerph-16-02624-t003:** General characteristics of selected studies.

Characteristics	Number (*n* = 34)	References
Publication year		
2011–2017	23	[14,15,16,17,18,19,20,21,22,23,24,25,32,33,34,35,36,37,38,41,43,44,46]
2000–2010	7	[26,27,28,29,42,45,47]
1985–1999	4	[30,31,39,40]
Type of paper		
Research article	27	[14,15,16,17,18,19,20,21,22,23,24,25,26,27,28,29,30,31,32,33,34,35,36,37,38,39,40]
Technical report	3	[43,44,47]
Master’s thesis	1	[42]
Undergraduate thesis	1	[45]
Conference paper	1	[46]
Discussion paper	1	[41]
Health topic		
Mosquito-borne diseases	18	[17,18,19,20,23,27,33,34,35,37,38,41,42,43,44,46,47]
Water-related diseases	9	[18,21,26,30,31,39,40,46,47]
Respiratory diseases	8	[15,16,22,24,25,36,45,46]
Other communicable diseases	3	[28,29,47]
Non-communicable diseases	4	[14,15,22,36]
Malnutrition	1	[32]
Study sites		
City-level	16	[14,15,16,18,19,22,29,30,31,33,34,35,36,38,39,40]
Province-level	7	[24,25,28,32,42,44,45]
Region-level	5	[16,27,37,44,47]
Country-level	8	[17,20,21,23,26,41,43,46]
First author affiliation		
Philippines	20	[19,26,27,28,29,30,31,32,33,34,35,37,38,40,41,43,44,47]
Japan	6	[14,15,16,18,22,36]
USA	3	[20,23,39]
Australia	3	[21,24,25]
Canada	1	[42]
Saudi Arabia	1	[17]
Funding source		
Foreign-based	18	[14,15,16,18,20,21,22,23,24,25,26,28,30,31,36,39,42,44]
Philippine-based	2	[19,39]
No statement of funding	15	[17,27,29,32,33,34,35,37,38,40,41,43,45,46,47]

**Table 4 ijerph-16-02624-t004:** Methodological characteristics of selected studies.

Characteristics	Number (*n* = 34)	References
Study design		
Time-series analysis	27	[14,15,16,18,19,20,22,25,27,28,31,32,33,34,35,36,37,38,39,40,41,42,43,44,45,46,47]
Cross-sectional	3	[26,29,30]
Process-based modeling	2	[17,24]
Bayesian modeling	1	[21]
Fuzzy association rule mining	1	[23]
Case-crossover	1	[25]
Exposure variables		
Meteorological parameters	27	[16,17,18,19,20,21,23,24,25,27,28,31,32,33,34,35,37,38,39,40,41,42,43,44,45,46,47]
Extreme weather events	5	[14,15,22,23,36]
Atmosphere-ocean interactions	3	[20,23,41]
Wet/dry seasons	3	[26,29,30]
Outcome variables		
Surveillance data	15	[16,17,18,19,20,23,27,37,38,41,43,44,45,46,47]
Hospital admissions	9	[24,25,28,29,33,34,35,39,42]
Vital statistics data	4	[14,15,22,36]
National surveys	3	[21,26,32]
Community-based	2	[19,40]
School-based	1	[31]
Outpatient-based	1	[30]

**Table 5 ijerph-16-02624-t005:** Research themes of selected studies.

Themes	Number (*n* = 34)	References
Ecosystem-Mediated Impacts of Climate Change on Health Outcomes	23	[17,18,19,20,21,23,26,27,30,31,33,34,35,37,38,39,40,41,42,43,44,46,47]
Direct Impacts of Climate and Weather on Health	11	[14,15,16,24,25,28,29,36,45,46,47]
Vulnerability to Disease and Injury Due to Climate Variability and Climate Change	4	[15,22,36,42]
Adaptation to Protect Health	4	[23,31,37,44]
Health Impacts Heavily Mediated through Human Institutions	1	[32]
Co-Benefits	0	-

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
