# Peer review of "Scoping Review of Climate Change and Health Research in the Philippines: A Complementary Tool in Research Agenda-Setting"

_ijerph, 2019, doi:10.3390/ijerph16142624_

Round 1

Reviewer 1 Report

Review for Manuscript ID: ijerph-494928 ‘Scoping review of climate and health research in the Philippines:
A complementary tool in research agenda-setting’

Overall comments:

This is a really interesting and important paper. However, it is really difficult to bring together the research findings from different research designs, specifically quant versus qual versus mixed methods. I think the paper and message from the paper would be much better articulated and more robust if the authors focused on reviewing just the quant papers. I comment more on this below.

Given the need to focus the paper more, I am suggesting major revisions and I happy to review paper again after revisions have been completed. 

Comments:

Introduction

Scoping reviews are well established, I think that the authors should focus on the topic of the scoping review as the first part of the introduction. So why is health and climate change important, what aspects, and then why the Philippines. The topic is the area of interest, start the paper from this perspective.

The same is true of the abstract, in the first sentence or 2 explain why climate change and health and why is it important in the Philippines. 

Then note the scoping reviews ability to act as an agenda setter and note the aim of the paper.

Materials and methods

I think it is important to note in this section why it is so important to include grey literature as many health research reviews only focus on journal articles. Specify why you chose to include grey literature (which I agree with).

Otherwise this is a very clear section.

Results

Can you specify in the results section of the 757 publications identified how many were journal articles and how many were grey literature?

ecologic study design? I think the authors mean ecological research design

It is really difficult to bring together the research findings from different research designs, specifically quant versus qual versus mixed methods. I think the paper and message from the paper would be much better articulated and more robust if the authors focused on reviewing just the quant papers. As it stands the results of t equal papers are only presented in a very quantitative manner which does not do justice to the deeper findings that qual research uncovers.

Discussion

The discussion section needs to be restructured in line with the aims implied in the title of the research. Please present a bullet point overview of the main topics identified for agenda setting and discuss what is required in the Philippines to meet these requirements. 

Then present the limitations/overview of the review.

Then focus on usefulness of a scoping review for agenda setting work.  

Author Response

Response to Reviewer 1:

Point 1: ...it is really difficult to bring together the research findings from different research designs, specifically quant versus qual versus mixed methods. I think the paper and message from the paper would be much better articulated and more robust if the authors focused on reviewing just the quant papers. 

Response 1: We revised as suggested and only included quantitative studies in the manuscript. Because of this, the results section was overhauled. See the results section at pages 4-10. 

Point 2: I think that the authors should focus on the topic of the scoping review as the first part of the introduction. So why is health and climate change important, what aspects, and then why the Philippines. The topic is the area of interest, start the paper from this perspective.

The same is true of the abstract, in the first sentence or 2 explain why climate change and health and why is it important in the Philippines. 

Then note the scoping reviews ability to act as an agenda setter and note the aim of the paper.

Response 2: We revised the abstract and introduction by following the suggested structure. Se pages 1-2

Point 3: Specify why you chose to include grey literature (which I agree with).

Response 3: Grey literature were searched to expand the possibly small number of published articles in scientific journals. See page 2 last paragraph

Point 4: Can you specify in the results section of the 757 publications identified how many were journal articles and how many were grey literature?

Response 4: 726 articles and 31 grey literature. See page 4.

Point 5: ecologic study design? I think the authors mean ecological research design

Response 5: Changed the categories of study design. See page 7.

Point 6: The discussion section needs to be restructured in line with the aims implied in the title of the research. Please present a bullet point overview of the main topics identified for agenda setting and discuss what is required in the Philippines to meet these requirements. Then present the limitations/overview of the review. Then focus on usefulness of a scoping review for agenda setting work.  

Response 6: Fully revised discussion section by following the suggested structure. See pages 10-12.

Reviewer 2 Report

Overall, this scoping review is a good start towards identifying gaps in research to help build a program for climate change and health research in the Philippines. The framework the authors use will to couch the review in research themes needed to address health issues regarding climate impacts. While the content and structure of the review are solid, there are some points that need to be clarified.

One major point is the issue of terminology. It is clear from the title of the manuscript and from the search strategy that this review is about climate change and health outcomes. I am concerned over the use of the term ‘climate/weather’ to describe the exposures. There are obviously differences in the definition of weather and climate and by using these terms interchangeably, it reduces the importance in the discussion of climate change hazards both acute and long-term on health.

I believe the authors should be clear in their terminology that this manuscript is focused on climate change hazards and not weather; or they should specify extreme weather events associated or linked to climate change. Part of defining and communication the impacts of climate change is to both make the association to extreme weather events, but not to say that climate and weather are the same thing.

In relation to the previous comment on terminology, the authors should be careful to define what they mean by seasonality, weather variables, and climate variables.

For example in North America and Europe seasonality means one thing (Fall, Winter, Spring, Summer) while in South Asia and the Philippines, seasonality may refer to wet and dry seasons with minimal temperature changes. Similarly, for weather variables, I would use the term meteorological variables to indicate temperature and rainfall. The authors also use the term climatic variables to describe ONI’s – again I would not use the term climatic variables and these are more so climate indicators or climate phenomenon.

In these cases word choice and definitions do matter and the authors should be specific in their use of terminology especially if their priorities are to build more capacity for climate change and health research in the Philippines.

Line 193 – Instead of ‘Results or findings’, just keep it as Findings.

Lines 208-210 – How are extreme temperatures defined as opposed to heat waves?  

Lines 248-252 – While this ADB report is from 2011, it is incorrect to say that the health sector does not contribute much to greenhouse gases. A quick review of more recent ABD reports conflicts with this statement. The authors should either remove this or identify updated reports/literature which discuss the current state of GHG emission in the Philippines.

Author Response

Responses to Reviewer 2

Point 1: One major point is the issue of terminology. It is clear from the title of the manuscript and from the search strategy that this review is about climate change and health outcomes. I am concerned over the use of the term ‘climate/weather’ to describe the exposures. There are obviously differences in the definition of weather and climate and by using these terms interchangeably, it reduces the importance in the discussion of climate change hazards both acute and long-term on health.

I believe the authors should be clear in their terminology that this manuscript is focused on climate change hazards and not weather; or they should specify extreme weather events associated or linked to climate change. Part of defining and communication the impacts of climate change is to both make the association to extreme weather events, but not to say that climate and weather are the same thing.

In relation to the previous comment on terminology, the authors should be careful to define what they mean by seasonality, weather variables, and climate variables.

For example in North America and Europe seasonality means one thing (Fall, Winter, Spring, Summer) while in South Asia and the Philippines, seasonality may refer to wet and dry seasons with minimal temperature changes. Similarly, for weather variables, I would use the term meteorological variables to indicate temperature and rainfall. The authors also use the term climatic variables to describe ONI’s – again I would not use the term climatic variables and these are more so climate indicators or climate phenomenon.

In these cases word choice and definitions do matter and the authors should be specific in their use of terminology especially if their priorities are to build more capacity for climate change and health research in the Philippines.

Response 1: Chose "Climate Factors" as umbrella term comprising meteorological parameters (i.e. temperature, rainfall, relative humidity etc.), extreme weather events (high temperature, heatwaves, typhoons), and atmosphere-ocean interaction phenomena (ENSO and sea surface temperature). See pages 3-4 and 7 (3rd paragraph).

Point 2: Line 193 – Instead of ‘Results or findings’, just keep it as Findings.

Response 2: Used "Findings" in the subheading at page 8.

Point 3: Lines 208-210 – How are extreme temperatures defined as opposed to heat waves?  

Response 3: Extreme temperatures were based on 95th and 99th percentile of temperature, and heatwaves are consecutive days of extreme temperature. See page 7 paragraph 3, and page 9 paragraph 3.

Point 4: Lines 248-252 – While this ADB report is from 2011, it is incorrect to say that the health sector does not contribute much to greenhouse gases. A quick review of more recent ABD reports conflicts with this statement. The authors should either remove this or identify updated reports/literature which discuss the current state of GHG emission in the Philippines.

Response 4: Removed the abovementioned statement from the ADB report.

Round 2

Reviewer 1 Report

This is a much improved manuscript. I would like to commend the authors for addressing my concerns. I am happy to accept the paper but recommend that the paper is read/edited by an english language specialist.